# Suppression of the alpha, delta, and omicron variants of SARS-Cov-2 in Taiwan

Hsiao-Hui Tsou[1,2]*, Fang-Jing Lee[1], Shiow-Ing Wu[1], Byron Fan[3], Hsiao-Yu Wu[1], Yu-Hsuan Lin[1,4,5,6], Ya-Ting Hsu[1], Chieh Cheng[1], Yu-Chieh Cheng[1], Wei-Ming Jiang[1], Hung-Yi Chiou[1,7,8], Wei J. Chen[9,10], Chao A. Hsiung[1], Pau-Chung Chen[11,12,13,14], Huey-Kang Sytwu[15]

1 Institute of Population Health Sciences, National Health Research Institutes, Zhunan, Miaoli County, Taiwan, 2 Graduate Institute of Biostatistics, College of Public Health, China Medical University, Taichung, Taiwan, 3 Brown University, Providence, Rhode Island, United States of America, 4 Department of Psychiatry, National Taiwan University Hospital, Taipei, Taiwan, 5 Department of Psychiatry, College of Medicine, National Taiwan University, Taipei, Taiwan, 6 Institute of Health Behaviors and Community Sciences, College of Public Health, National Taiwan University, Taipei, Taiwan, 7 School of Public Health, College of Public Health, Taipei Medical University, Taipei, Taiwan, 8 Master's Program in Applied Epidemiology, College of Public Health, Taipei Medical University, Taipei, Taiwan, 9 Center for Neuropsychiatric Research, National Health Research Institutes, Zhunan, Miaoli County, Taiwan, 10 Institute of Epidemiology and Preventive Medicine, College of Public Health, National Taiwan University, Taipei, Taiwan, 11 National Institute of Environmental Health Sciences, National Health Research Institutes, Zhunan, Miaoli County, Taiwan, 12 Institute of Environmental and Occupational Health Sciences, National Taiwan University College of Public Health, Taipei, Taiwan, 13 Department of Public Health, National Taiwan University College of Public Health, Taipei, Taiwan, 14 Department of Environmental and Occupational Medicine, National Taiwan University Hospital and National Taiwan University College of Medicine, Taipei, Taiwan, 15 National Institute of Infectious Diseases and Vaccinology, National Health Research Institutes, Zhunan, Miaoli County, Taiwan

* tsouhh@nhri.org.tw

## Abstract

### Background

Taiwan was a coronavirus disease 2019 (COVID-19) outlier, with an extraordinarily long transmission-free record: 253 days without locally transmitted infections while the rest of the world battled wave after wave of infection. The appearance of the alpha variant in May 2021, closely followed by the delta variant, disrupted this transmission-free streak. However, despite low vaccination coverage (<1%), outbreaks were well-controlled.

### Methods

This study analyzed the time to border closure and conducted one-sample t test to compare between Taiwan and Non-Taiwan countries prior to vaccine introduction. The study also collected case data to observe the dynamics of omicron transmission. Time-varying reproduction number, $R_t$, was calculated and was used to reflect infection impact at specified time points and model trends of future incidence.

### Results

The study analyzed and compare the time to border closure in Taiwan and non-Taiwan countries. The mean times to any border closure from the first domestic case within each

**Data Availability Statement:** The minimal data set is available at the GitHub repository [https://github.com/ChiehCheng/COVID-19-in-Taiwan]. The population flow data are third party data. These data are not owned or collected by the authors.

Anyone who has an interest in this data set could contact the following consulting webpage of Far EasTone Telecommunications for more information. [https://enterprise.fetnet.net/content/ebu/tw/product/IOT/IOT_L3/smart-city-big-data.html].

**Funding:** (1)Initials of the authors who received each award:H.H.T. (2)Grant numbers awarded to each author:grant PH-111-PP-02, PH-111-GP-02, PH-112-PP-02, and PH-112-GP-02 (3)The full name of each funder:National Health Research Institutes (4)URL of each funder website:https://www.nhri.edu.tw/ (5)Did the sponsors or funders play any role in the study design, data collection and analysis, decision to publish, or preparation of the manuscript? NO - Include this sentence at the end of your statement: The funders had no role in study design, data collection and analysis, decision to publish, or preparation of the manuscript.

**Competing interests:** The authors have declared that no competing interests exist.

country were -21 and 5.98 days, respectively (P < .0001). The Taiwanese government invested in quick and effective contact tracing with a precise quarantine strategy in lieu of a strict lockdown. Residents followed recommendations based on self-discipline and unity. The self-discipline in action is evidenced in Google mobility reports. The central and local governments worked together to enact non-pharmaceutical interventions (NPIs), including universal masking, social distancing, limited unnecessary gatherings, systematic contact tracing, and enhanced quarantine measures. The people cooperated actively with pandemic-prevention regulations, including vaccination and preventive NPIs.

## Conclusions

This article describes four key factors underlying Taiwan's success in controlling COVID-19 transmission: quick responses; effective control measures with new technologies and rolling knowledge updates; unity and cooperation among Taiwanese government agencies, private companies and organizations, and individual citizens; and Taiwanese self-discipline.

## Introduction

Taiwan has been described as a coronavirus disease 2019 (COVID-19) outlier due to its maintaining the longest continuous transmission-free record in the world: 253 days without locally transmitted infection beginning April 12, 2020 [1–3]. As of December 31, 2020, Taiwan had 799 confirmed COVID-19 cases and only 7 deaths among its ~23.8 million citizens [4]. Schools were open, restaurants were bustling, concerts were playing, and theaters were crowded. Taiwan was the only country that experienced economic growth in 2020 [5].

The alpha variant of severe acute respiratory syndrome coronavirus 2 (SARS-CoV-2, the virus that causes COVID-19) was first detected in Taiwan in January 2021 and the resulting cluster of 21 local infections was cleared within 27 days [6, 7]. Life then returned to normal and months passed without any additional local infections. Large-scale events and family gatherings became increasingly common and people relaxed. This enviable life was ended in May 2021 by a COVID-19 outbreak triggered by re-introduction of the alpha variant, causing more than 14,400 new local cases by mid-August.

Although Taiwan's vaccination coverage rate was <1% when the outbreak began, local cases still dropped precipitously from May to August (Fig 1). In June 2021, the delta variant was identified in Southern Taiwan and effective measures were taken immediately. Transmission was contained within 10 days [8] and there were zero new local cases by mid-August [4].

The first community-spread cases of the omicron variant in Taiwan were detected in January 2022 [9]. Between January and early March there was only a modest rise in omicron variant cases in Taiwan; the highest daily case number during this period was 63, and by March 2 there were only 5 daily cases [4]. The epidemiological trajectories of each new variant outbreak in Taiwan were distinct from those elsewhere in the world. This article explores the factors that allowed Taiwan to suppress COVID-19 outbreaks following introductions of the alpha, delta, and omicron variants of SARS-CoV-2.

## Methods

### Geographic comparison

We analyzed the time to border closure and conducted one-sample t-test to compare between Taiwan and Non-Taiwan countries prior to vaccine introduction. Data about public health

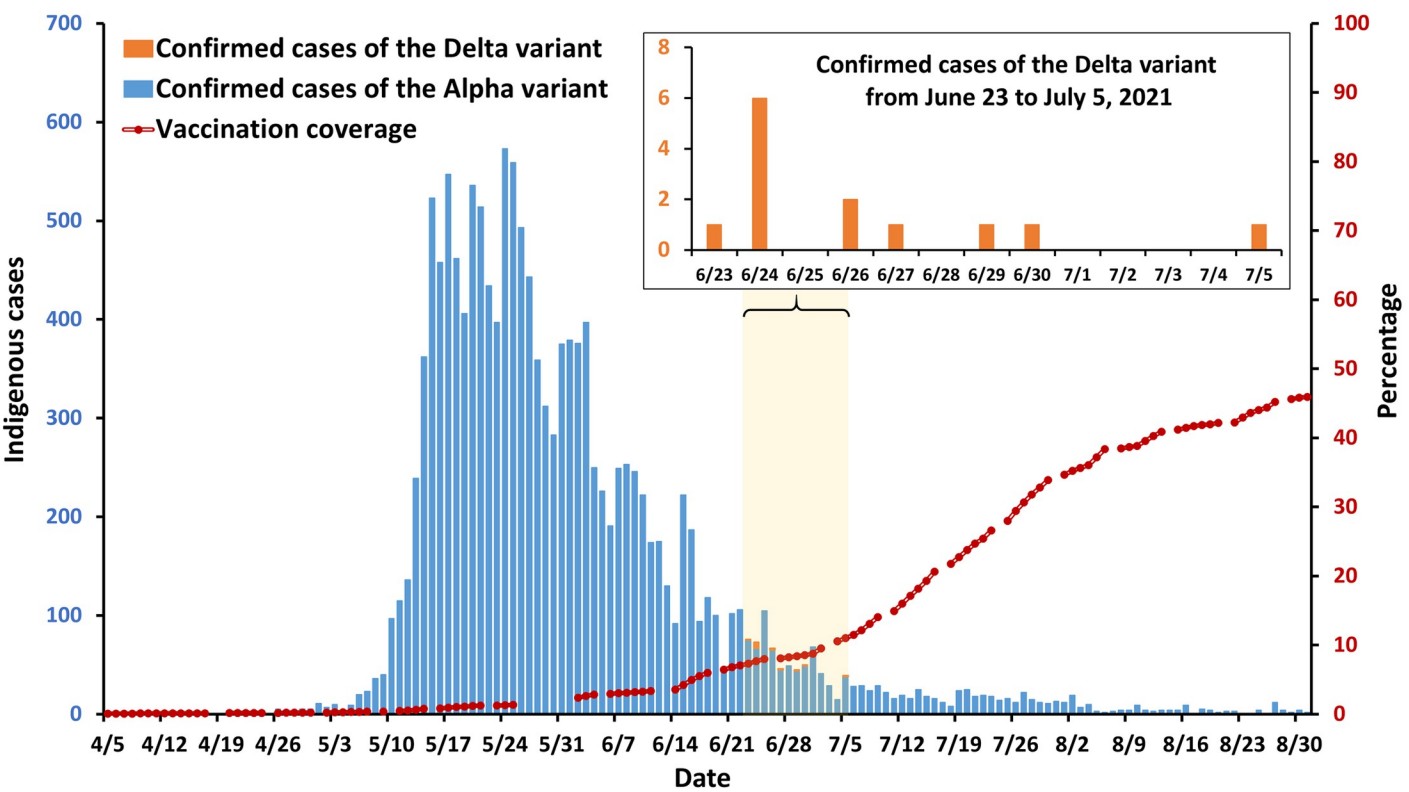

**Fig 1. Confirmed cases and vaccination coverage in Taiwan from April 5 to August 31, 2021.** Confirmed cases include community-spread cases attributed to each variant. Vaccination coverage is defined as the ratio of the total number of COVID-19 vaccination doses administered in Taiwan to the total population of Taiwan.

policies for each country were captured through the Oxford COVID-19 Government Response Tracker [10]. We collected the variable of "international travel controls (C8)" from the Oxford dataset to determine the time (in days) between the first reported case to implementation of border closure in each country. We identified the initial time of policy implementation as the time when C8 first becomes nonzero. (The information of border control measures is in supporting information 1 of S1 Appendix) Similarly, the first reported cases in each country are based on the COVID-19 dataset from Our World in Data [11]. For example, the first reported case in China is 2019/12/31, border control measures in Taiwan were implemented on 2020/01/01, and the first case in Taiwan is 2021/01/22. The time to any border closure from first reported case in China and Taiwan are 1 and -21 day, respectively. Forty- nine non-Taiwan countries were selected from the report "A comprehensive evaluation of COVID-19 policies and outcomes in 50 countries and territories" [12]. Statistical analyses were conducted via SAS software (version 9.4); *P* values were two-tailed with 0.05 significance level.

## Evaluation of suppression by time-varying reproduction number

We collected case data from November 26, 2021 to February 22, 2022, a time period encompassing the introduction of indigenous omicron variant infections, to observe the dynamics of omicron transmission. Time-varying reproduction number, $R_t$ was calculated as described by Wallinga and Teunis [13] and by Cori et al. [14], and was used to reflect infection impact at specified time points and to model trends of future incidence [15]. $R_t$ is the expected number of new infections caused by each infectious individual. The analyses were conducted in R

(version 3.4.2) and RStudio (version 1.1.383); $R_t$ was evaluated with EpiEstim (https://rdrr.io/cran/EpiEstim/). Please see the supporting Information 2 of S1 Appendix.

### Ethics statement

The institutional review board of the National Health Research Institutes approved this study in the form of written consent (approval number: EC1091110-E).

## Results

### Taiwan's responses to the alpha, delta, and omicron variants

Taiwan made policies toward the goal of COVID-Zero during the period of alpha and delta variants, including strict border control measures, contact tracing, and home quarantine for 14 days. It is quite remarkable that Taiwan was able to reduce the number of new local cases to zero in a span of 108 days despite invasions of the alpha and delta variants. In addition, under the implementation of the vaccine policy by the government, vaccination coverage in Taiwan has gradually increased. The research institute in Taiwan has also continued to conduct research on the screening strategy of border opening and the impact of vaccine efficacy and coverage on the risk of infection.

The high transmissibility of the Omicron variant is a major cause of global concern. Since the appearance of Omicron, it has quickly replaced Delta as the dominant strain worldwide [16]. Although Taiwan encountered the omicron epidemic in early January 2022, the effective reproductive number dropped rapidly from 2.5 to less than 1 from January by the end of February (Fig 2). A plot of mean $R_t$ values during the aforementioned time period, shown with 95% confidence interval values in Fig 2 demonstrates rapid suppression of a potential omicron variant outbreak. Taiwan's control of trans-mission can be attributed to four factors, elaborated below. According to Our World in Data, by October 24, 2022, Taiwan's case fatality rate was 0.17%, ranking sixth among Organization for Economic Cooperation and Development (OECD) countries (Fig 3) [11, 17].

**Factor 1: Quick response.** More than many other countries, Taiwan prioritized swiftness in its pandemic response. On December 31, 2019, Taiwanese authorities began screening passengers on flights from Wuhan, China, where the virus was first discovered [18]. Tsou et al. provided a comprehensive evaluation of COVID-19 policies and out-comes in 50 countries and territories, and reported that border control policies were implemented earlier in Asian countries than in non-Asian countries [12]. We further analyzed Tsou et al.'s data to compare Taiwan to all other countries and found that the mean time to any border closure from the first reported case in China were 1 and 49.76 days, respectively (P < .0001), and the mean times to any border closure from the first domestic case within each country were -21 and 5.98 days, respectively (P < .0001) (Table 1). We also compared the epidemic outcomes-related performance of Taiwan and other island countries, please see the supporting information 1 of S1 Appendix.

From January 20 to February 24, 2020, the Taiwanese government implemented at least 124 action items, including border control, case identification, contact tracing, and quarantine policies [19, 20]. In January 2020, Taiwan's Central Epidemic Command Center began holding daily press briefings and regularly releasing news about the epidemic to educate the public while combating misinformation [21]. In accordance with the Communicable Disease Control Act, the Taiwanese CDC classified COVID-19 as a Category 5 communicable disease on January 15, 2020, thereby strengthening surveillance and containment [22]. This act urged medical institutions and the public to remain vigilant and take preventive measures to reduce transmission risk, violators were fined or forcibly quarantined.

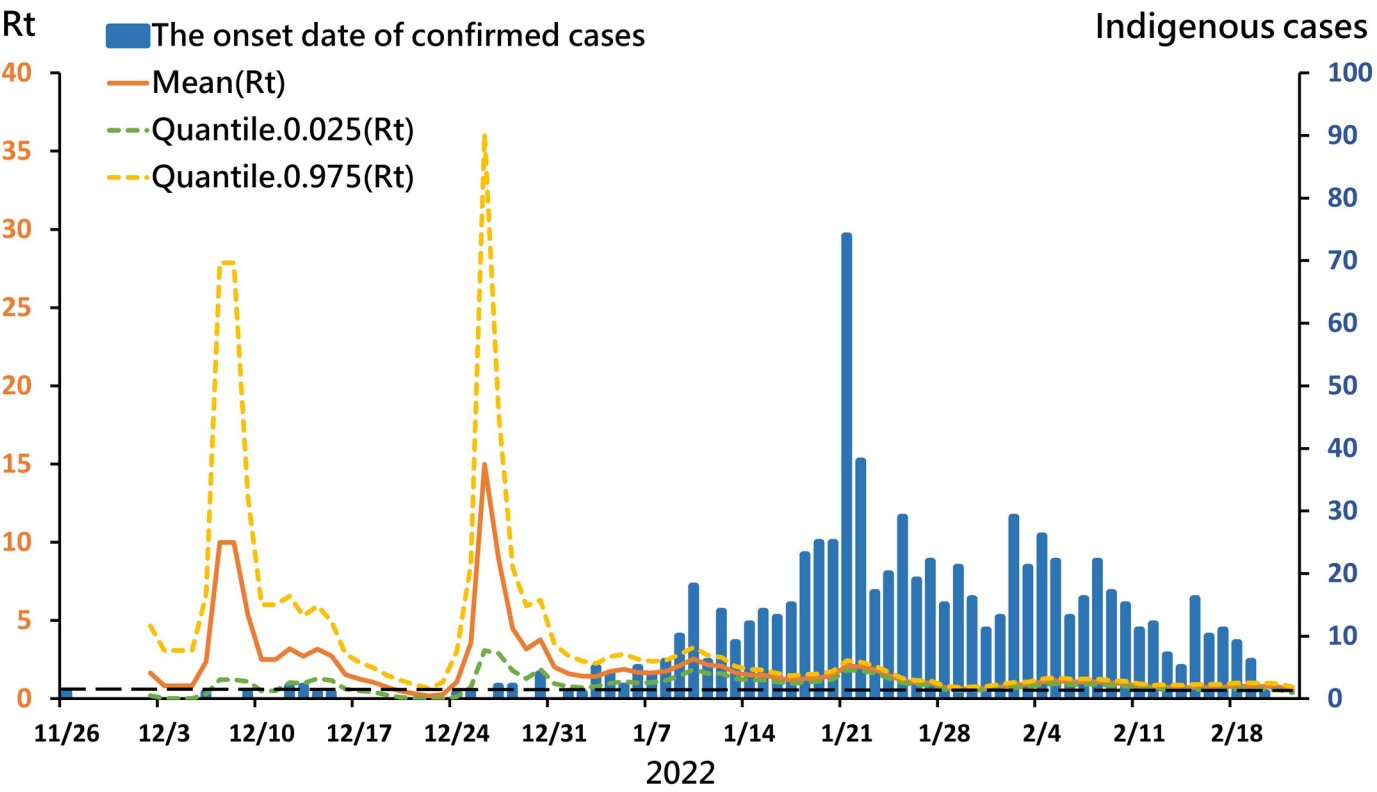

**Fig 2. Confirmed cases of SARS-CoV-2 omicron variant infection and effective reproduction number, Rt, in Taiwan from November 26, 2021 to February 22, 2022.** Confirmed indigenous cases are shown as blue bars by date (x-axis) and correspond to the right-sided y-axis. A curve of SARS-CoV-2 omicron variant Rt (left y-axis) over time (x-axis) is shown in orange with the lower (green) and upper (yellow) bounds of the associated 95% confidence interval. Note that despite the high transmissibility of the variant, case growth was suppressed rapidly and effectively.

**Factor 2: Effective control measures and precise quarantine strategy.** Contact tracing is an essential strategy to curb COVID-19 transmission, especially in the early stages of an outbreak. This approach to disease control has several clear advantages. It not only provides a better understanding of the potential transmissibility of COVID-19, but also aids in the formulation of appropriate control strategies. Additionally, identifying the infection routes among healthcare workers enables the provision of suitable personal protective equipment (PPE) [23]. However, a prevalent issue that arises in the era of technology-based tracing is the privacy of individual citizens. Participation in contact tracing programs must be voluntary or self-reported. Thus, many countries struggled to implement this basic public health procedure. In England, contact-tracers failed to contact one in eight people who tested positive for COVID-19, and in 18% of confirmed cases, the infected people did not provide close-contact details [24]. In some US states, the majority of people who tested positive did not provide contact details [24].

The findings of a Taiwanese report on the transmission dynamics of COVID-19 in its initial 100 confirmed cases highlighted the need for accurate and comprehensive contact tracing and testing [23]. Furthermore, a modeling study assessing the potential effectiveness of control strategies for outbreak containment showed that isolating symptomatic patients may be insufficient and that isolation of all at-risk people would be an effective alternative [25].

The Taiwanese government invested in quick and effective contact tracing with a precise quarantine strategy [22]. With the assistance of patients and morally motivated contact-

## Case fatality rate (%)

**Fig 3. The comparison of Taiwan and select OECD countries for COVID-19 case fatality rate.**

tracers, dozens to hundreds of close contacts were identified for each individual case. From 2020 through March 2022, over 99% of close contacts complied with the 14-day mandated quarantines [22].

In response to the rapid spread of delta and omicron variants worldwide, Taiwan's Central Epidemic Command Center implemented longstanding measures, such as quarantine and isolation, to contain transmission [26]. People who violated quarantine regulations were fined, and people who complied were compensated; quarantine measures and compensation details are provided in the S1 Appendix. A modeling analysis using case data from Taiwan concluded that population-based interventions likely played a major role in Taiwan's initial elimination efforts, and case-based interventions alone were not sufficient to control the epidemic [15].

**Table 1. Comparison of the time to border closure among geographic regions.**

| Time to any border closure from first reported case in China (days)[1] | N | Mean (SE) | P value[2] |
|---|---|---|---|
| Taiwan | 1 | 1 | < .0001 |
| Non- Taiwan[3] | 49 | 49.76 (4.26) | |
| **Time to any border closure from first case in reference country (days)[4]** | | | |
| Taiwan | 1 | -21 | < .0001 |
| Non- Taiwan[3] | 49 | 5.98 (4.24) | |

[1] The first reported case in China was 31 December 2019.

[2] One-Sample t-test

[3] Please refer to the report "A comprehensive evaluation of COVID-19 policies and outcomes in 50 countries and territories" [12]

[4] Reference country is each index country

Thus, mandated interventions (contact tracing, quarantine, border control) are important and effective relative to non-mandated interventions (mask use, social distancing).

**Factor 3: Individual self-discipline.** In lieu of a strict lockdown, residents of Taiwan were encouraged strongly, but not mandated, to follow recommendations based on self-discipline, unity, and patience. Self-discipline was evidenced in Google mobility reports [27]: crowds in retail sites, recreation sites, parks, transit stations, and workplaces declined significantly after the Taipei City and New Taipei City Level- 3 alerts (Fig 4). Notably, crowds at transit stations were halved and normally lively streets became deserted, making it clear that people chose not to go out and gather in crowds.

During the Level-3 alert, the Taiwanese government shifted in-person classes to distance learning. Additionally, many companies implemented policies for off-site work to support physical distancing, reducing workplace inhabitance by 30% overall, with a sharp 70% decrease in mid-June. There was a modest uptick (10–20%) in residential gatherings during this same period.

After the Level-3 COVID-19 alert was implemented in Taiwan on 15 May 2021, the average number of daily trips dropped sharply, especially in urban areas (Fig 5). In Taipei city, there was a 52% reduction in daily trips; for all cities, there was an average reduction of 28%.

**Factor 4: Unity and cooperation.** There has been strong unity and cooperation among government agencies and the public in Taiwan. According to an Imperial College London YouGov Covid-19 Behavior Tracker survey administered in 28 countries [28], 32.3% of people in those countries reported greater solidarity since the pandemic began; in Taiwan, 49.9% of people felt more united (Fig 6a). In other countries surveyed, only 19.7% of people thought that the government was handling the issue very well; in Taiwan 37.6% of people held this view (Fig 6b). The average opinion in other counties was that 24.6% of people had a lot of confidence in the government's response to the pandemic; in Taiwan 30.6% of people held this view (Fig 6c) (please see details in the S1 Appendix).

In accordance with Level-3 epidemic alert measures (details on these measures are described in the S1 Appendix), Taiwan's central and local governments worked together to enact non-pharmaceutical interventions (NPIs) including universal masking, social distancing, limiting unnecessary activities and gatherings, systematic contact tracing, and enhanced quarantine measures [29]. There was good cooperation among private organizations to facilitate vaccine dissemination, as well as cooperation among citizens to get vaccinated [30]. The public cooperated actively with pandemic-prevention regulations, including vaccination and preventive NPIs, and even the worst outbreak was under control within 3 months.

## COVID-19 vaccination

Taiwan experienced its greatest COVID-19 case load thus far in the summer of 2021 while still having insufficient vaccination (Fig 1). Given the constantly evolving SARS-CoV-2 virus, only a large-scale vaccination campaign can resolve the ongoing COVID-19 pandemic completely. Detailed information on vaccination coverage, vaccination support, and vaccine donation is provided in the S1 Appendix.

## Delta variant hit southern Taiwan

The delta (B.1.617.2) variant was first detected in Taiwan on June 23, 2021 (please see details on the delta variant in the S1 Appendix) when the alpha variant was still dominant locally. The first cluster of delta variant infections was discovered in Fangshan Township, Pingtung

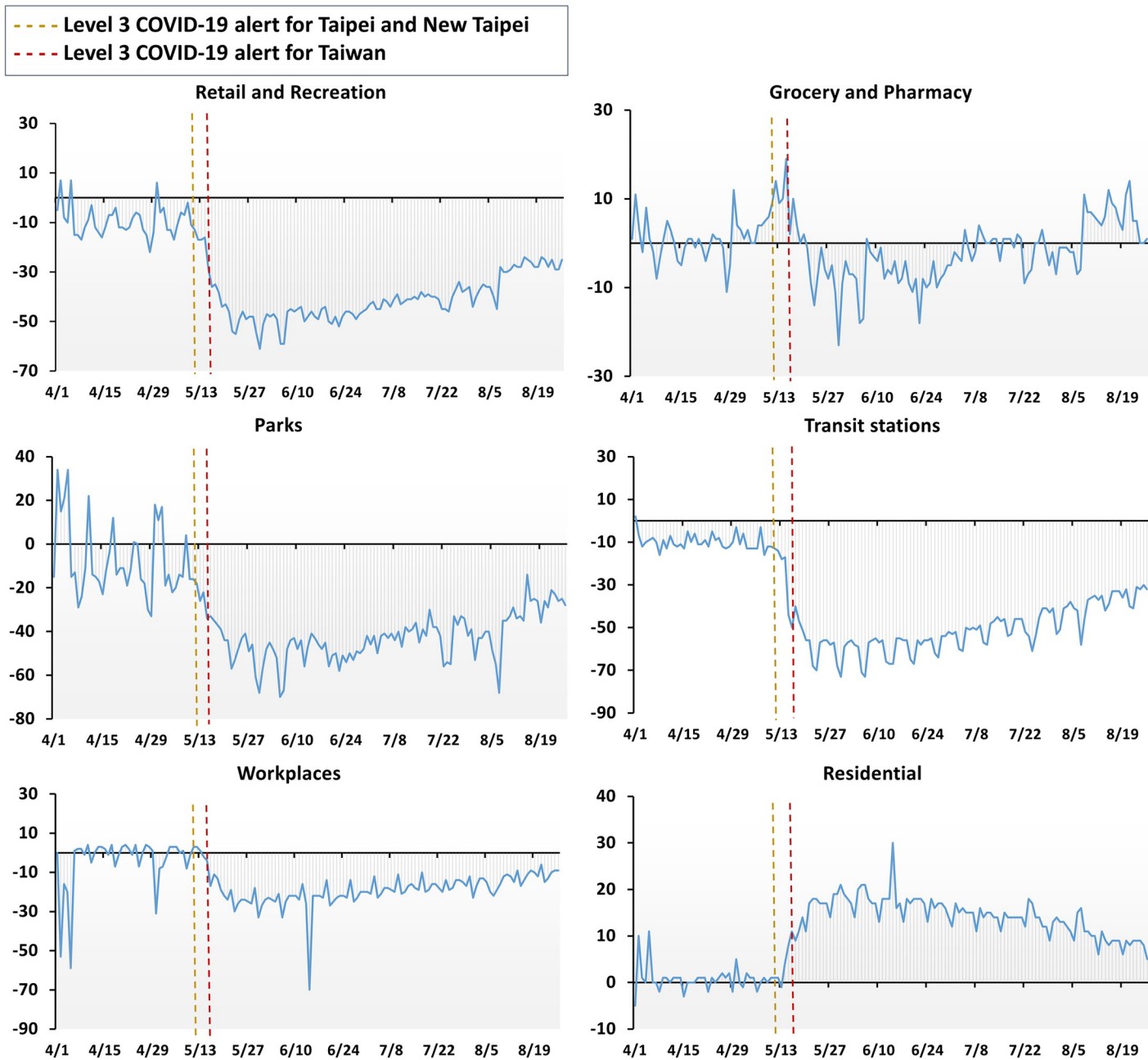

**Fig 4. Community mobility in Taiwan from April 1 to August 26, 2021.** The figure shows movement trends within Taiwan before/after the Taipei City and New Taipei City Level-3 alerts. The graphs illustrate how community members decreased their time in public spaces and increased their time in residential spaces after the alerts were issued. The x-axes show dates and y-axes represent the percent change from baseline. Baseline was defined for each day of the week as the median value from the 5-week period of January 3–February 6, 2020.

County, a small town in southern Taiwan; the delta variant then spread quickly to the neighboring town of Fangliao Township (Fig 7). Immediately, the government conducted a large-scale testing campaign of more than 14,000 residents to detect and isolate the virus. Working cooperatively, the central and local governments were able to get the delta variant outbreak under control within 10 days (Fig 1) [8].

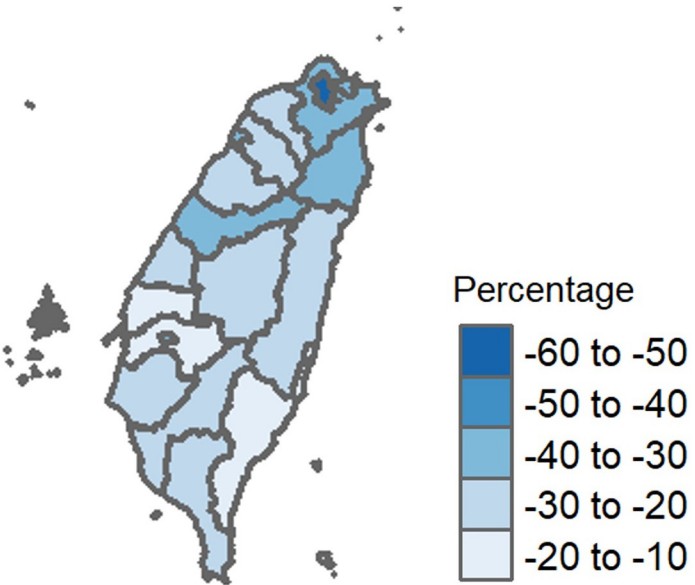

**Fig 5. The change in human mobility\* in Taiwan before and after the Level-3 COVID-19 alert issued on May 15, 2021.** The percent change in daily trips between pre- and post-alert periods. \*Human mobility: daily mobile phone data obtained from Far EasTone Telecommunications in Taiwan.

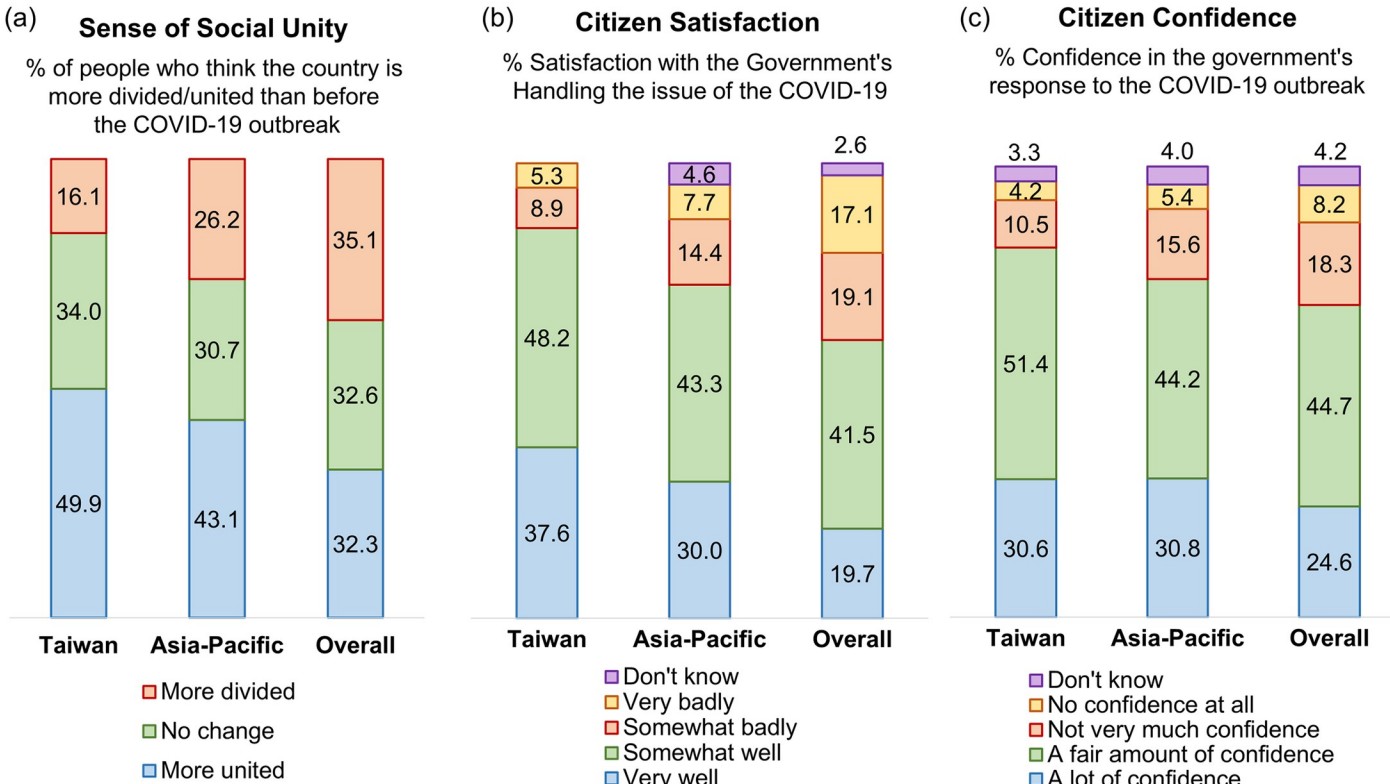

**Fig 6. Unity and cooperation sentiments during COVID-19 pandemic in Taiwan.** Data was processed from the Imperial College London YouGov Covid-19 Behavior Tracker, a survey administered in 28 countries in September 2020. Charts show the range of sentiments on (a) social unity, (b) government handling of the pandemic, and (c) citizen confidence in their government's response.

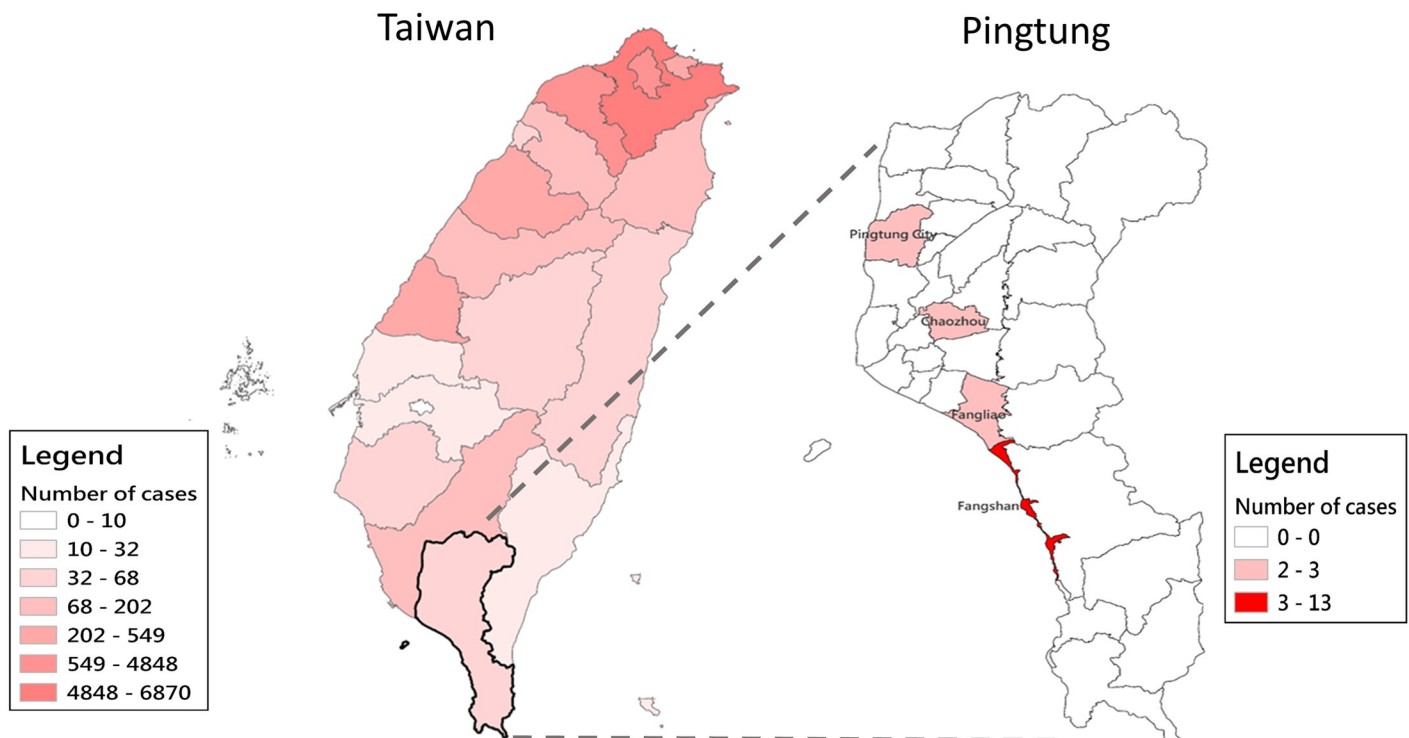

**Fig 7. COVID-19 outbreak in Taiwan from April 5 to August 31, 2021.** A delta variant infection cluster involving the Fangshan and Fangliao Townships in Pingtung County was discovered in June of 2021.

## Taiwan on high SARS-CoV-2 omicron variant alert

The omicron variant of SARS-CoV-2 (B.1.1.529) was first identified in South Africa before it proceeded to spread internationally. Its emergence was associated with rapid surges in case numbers and quickly became the predominant circulating strain [31–33]. The omicron variant is highly infectious and resistant to vaccine neutralization, but has been associated with less morbidity than previous variants [34, 35]. In early January 2022, Taiwan detected indigenous omicron coronavirus variant spreading in the community related to workers at the Taoyuan International Airport due to transmission from infected arriving passengers [9, 36]. In response, Taiwan's government ordered tighter controls, taking immediate action to prevent a sudden surge of cases [37, 38].

Taiwan remains on high alert for SARS-CoV-2 variants in 2022, despite actually only having 855 local omicron variant infections documented from January 1 to March 4, 2022 [4]. Numerous studies indicated that most cases of omicron are mild or asymptomatic [39–42], which hint at a preview of a future with endemic SARS-CoV-2. The world is gradually shifting towards restriction-free policies during the omicron period and Taiwan is following that pattern. Low hospitalization and death rates persist as people in Taiwan are going about their normal lives, as seen in Google mobility reports (2022 v. 2021 in Figure S1 of S1 Appendix) [43]. As a result, Taiwan's government has made efforts to encourage vaccination and made booster shots available in response to the omicron variant outbreak. At least 77.2% of people in Taiwan have received two COVID-19 vaccine doses and about 44.8% of Taiwanese residents had received their third vaccine dose as of March 7, 2022 [44].

In reference to the study of scientific and empirical data, the Taiwanese government announced that the quarantine duration for all arrivals would be reduced from 14 to 10 days starting on March 7, 2022 [45]. In April 2022, they also announced the implementation of key contact tracing starting April 25 and shortening of the home isolation period with a 3+4 plan starting April 26 [46]. Taiwan's goal during the omicron period was to reduce severe cases and manage mild cases, while moving toward a new normal lifestyle of coexisting with the coronavirus and effective epidemic control.

## Discussion

Because of Taiwan's close proximity to mainland China, there are large numbers of regular flights between Taiwan and the mainland. Consequently, Taiwan was predicted in early 2020 to be at second-highest risk of importing COVID-19 cases in the world [47]. However, protective measures enacted by the Taiwanese government protected Taiwan from severe outbreaks, with epidemic prevention and healthcare professionals working tirelessly on the front lines. The Taiwanese government setting operates the National Health Insurance (NHI) and Central Epidemic Command Center. The former ensures everyone has equal medical resources and helps the government check clinical records and allocate resources such as rapid tests and masks. The latter focuses on the country's epidemic safety. At the individual level, they improved people's confidence in government and knowledge through transparent public press conferences and public health education videos. At the public level, border control and strict quarantine measures were promulgated very early, and case tracking and contact tracing were implemented effectively throughout the pandemic [20].

Mask production was increased in preparation for potential outbreaks and a rationing system was implemented to enable people to order masks online and receive them at any National Health Insurance pharmacy [48]. When mask production was abundant, Taiwan was able to provide masks to other countries in need. Besides the government providing masks, Taiwan is "an island of masks" [49]. Different from the Western countries, sick people wearing masks in public is considered common and polite behavior to protect other people from being infected. This enhances the willingness and the number of people wearing masks.

Faced with the challenge of emerging variants of concern, the Taiwanese government expanded screening and testing, particularly in high-risk areas. Meanwhile, previously enacted transmission prevention measures—including timely border protection, aggressive contact tracing, and successful quarantine policies—continued to be implemented.

Both case-based interventions (e.g., contact tracing, quarantine, border control) and population-based interventions (e.g., use of masks, social distancing) may be important to control the spread of COVID-19 in Taiwan. It is necessary to not only have accurate, comprehensive contract tracing and testing, but also it is essential to identify and isolate symptomatic patients or all at-risk cases to control the outbreak [25, 50]. Although individual self-discipline is unable to be enforced by legal legislation while mandatory restrictions can, individual self-discipline would reinforce the possibility of interrupting transmission. In other words, either category of interventions alone was insufficient to control the epidemic. Taiwan's control strategy includes the combination of case-based and population-based interventions which was indicated effective and may be interpreted as a reason for Taiwan's control success [15].

When global vaccine supply was insufficient, the Taiwanese government cooperated with the private sector to maximize vaccine access (please see details in the S1 Appendix). When Taiwan faced a local outbreak in the summer of 2021, many countries donated vaccines to

Taiwan. Throughout the pandemic, the people of Taiwan have cooperated with public health laws and regulations and engaged in self-discipline. The common societal conformity led the Taiwanese to more immediately follow directives rather than hesitate and ask for the reason, which saves time during the epidemic [49], enabling NPIs to be highly effective and suppressing outbreaks due to emerging variants of concern.

In examining the economic impact of COVID-zero approach in Taiwan in the first two years of pandemic (2020 and 2021), we focus on gross domestic product (GDP) and unemployment rate. Data on GDP and unemployment rate were collected from Trading Economics [51, 52], while data on cumulative deaths per million people were collected from Our World in Data [11]. Acknowledging the emergence of the Omicron variant in late November 2021 [53], we limited our scope to the period before November 2021. Using the data available up to that point in time, we built scatter plots to illustrate the distribution of cumulative deaths per million people against GDP and unemployment rate. Figures of the supporting Information 9 of S1 Appendix illustrate that, in comparison to other countries, Taiwan's cumulative deaths per million people and economic indicators such as unemployment rate between December 2020 and November 2021 are considered good since they are notably low (Supplementary Figures S11 and S12 of S1 Appendix). Furthermore, an examination of the GDP data reveals that Taiwan's economy remained relatively stable in the first two years of pandemic (2020 and 2021), experiencing no significant fluctuations (Supplementary Figures S13 and S14 of S1 Appendix).

During the initial stages of the COVID-19 outbreak, particularly in the Alpha and Delta phases, Taiwan distinguished itself for its exceptional performance and resilience across various metrics. According to the Mental Health Foundation's 2020 biannual national mental health index survey, Taiwan achieved an impressive overall score of 83.3 [54]. Despite the challenges posed by the Omicron phase in 2023, Taiwan's 116th rank out of 194 countries points to a relatively low level of depression, with a depression rate of 3.59%. This rate is notably lower than that of Greece, which recorded the highest rate at 6.52% [55].

Furthermore, Taiwan's score of 83.8 on the 2023 country safety index by Numbeo solidifies its place as one of the top three safest countries, following only Qatar with a score of 85.7 and the United Arab Emirates with a score of 85.4 [56]. These achievements underscore the success of Taiwan's early Zero COVID-19 strategy. Importantly, the implementation of the COVID-zero approach during the initial two-year period (2020–2022) did not show any adverse psychological effects on the public's mental well-being or social psychological health, as evidenced by the survey findings [57].

This study has some limitations. First, it is commonly acknowledged that the susceptibility and transmissibility of the SARS-CoV-2 virus vary among countries. However, this study could not address interpersonal and geographical heterogeneity comprehensively, mainly because of limited data accessibility. This includes variations in predisposing conditions and compliance with epidemic control measures. Another constraint arises from the challenge of obtaining the vaccination status of those who are infected, making it challenging to assess the impact of vaccination on the Rt values.

## Conclusions

All available evidence indicates that Taiwan's successful response to COVID-19 has depended on multiple active factors rather than being a matter of passive luck. Key factors that contributed to Taiwan's resilience include rapid response, effective control strategies, self-discipline, and cooperation.

### Evidence for practice

- Taiwan has been a COVID-19 outlier with an extraordinarily long continuous virus-free record: 253 days without any local infections in 2020.

- The streak was disrupted by an alpha variant outbreak in mid-May 2021, and soon thereafter followed by a delta variant outbreak.

- Despite low vaccination coverage, locally transmitted cases in mid-May 2021 were managed effectively and the number of daily new cases fell from 535 to 6 in only three months.

- Key factors for Taiwan's success in controlling COVID-19 transmission: quick responses; effective control measures with new technologies and knowledge; unity and cooperation among government agencies, private companies and organizations, and individual citizens; and Taiwanese self-discipline.

## Supporting information

**S1 Appendix.**
(DOCX)

## Acknowledgments

The authors thank Dr. Kung-Yee Liang and Dr. Shu-Chen Kuo from National Health Research Institutes, Taiwan, for their valuable comments and support. The authors thank Environmental & GeoInformatic Technology Co. Ltd for their help in producing a figure. The de-identification mobile phone signaling data used in this study were collected and aggregated by Far EasTone Telecommunications in Taiwan.

## Author Contributions

**Conceptualization:** Hsiao-Hui Tsou, Fang-Jing Lee, Shiow-Ing Wu, Hsiao-Yu Wu, Yu-Hsuan Lin, Ya-Ting Hsu, Yu-Chieh Cheng, Chao A. Hsiung, Huey-Kang Sytwu.

**Data curation:** Hsiao-Hui Tsou, Fang-Jing Lee, Hsiao-Yu Wu, Ya-Ting Hsu, Chieh Cheng, Wei-Ming Jiang.

**Formal analysis:** Hsiao-Hui Tsou, Fang-Jing Lee, Shiow-Ing Wu, Hsiao-Yu Wu, Ya-Ting Hsu, Chieh Cheng, Yu-Chieh Cheng, Wei-Ming Jiang.

**Funding acquisition:** Shiow-Ing Wu, Hung-Yi Chiou, Wei J. Chen, Chao A. Hsiung, Pau-Chung Chen, Huey-Kang Sytwu.

**Investigation:** Hsiao-Hui Tsou, Fang-Jing Lee, Hsiao-Yu Wu, Ya-Ting Hsu, Chieh Cheng.

**Methodology:** Hsiao-Hui Tsou, Fang-Jing Lee, Shiow-Ing Wu, Yu-Chieh Cheng.

**Project administration:** Hsiao-Hui Tsou, Fang-Jing Lee, Shiow-Ing Wu.

**Resources:** Shiow-Ing Wu, Yu-Hsuan Lin, Hung-Yi Chiou, Wei J. Chen, Chao A. Hsiung, Pau-Chung Chen, Huey-Kang Sytwu.

**Validation:** Hsiao-Hui Tsou, Hsiao-Yu Wu, Ya-Ting Hsu, Chieh Cheng, Yu-Chieh Cheng, Wei-Ming Jiang.

**Writing – original draft:** Hsiao-Hui Tsou, Fang-Jing Lee, Shiow-Ing Wu, Byron Fan.

**Writing – review & editing:** Hsiao-Hui Tsou, Fang-Jing Lee, Shiow-Ing Wu, Byron Fan.

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
