## [Decision Letter · Decision Letter 0]

16 Oct 2023

PONE-D-23-29703Suppression of the alpha, delta, and omicron variants of SARS-Cov-2 in Taiwan.PLOS ONE

Dear Dr. Tsou,

Thank you for submitting your manuscript to PLOS ONE. After careful consideration, we feel that it has merit but does not fully meet PLOS ONE’s publication criteria as it currently stands. Therefore, we invite you to submit a revised version of the manuscript that addresses the points raised during the review process. Thank you for submitting your important manuscript to PLOS ONE.I think the comments raised by the 2 independent reviewers seem minor.Please submit your revised version soon.

We look forward to receiving your revised manuscript.

Kind regards,

Etsuro Ito, Ph.D.

Academic Editor

PLOS ONE

Journal Requirements:

2. Thank you for stating the following in your Competing Interests section: "All authors declare no conflicts of interest, including employment, consultancies, stock ownership, honoraria, paid expert testimony, patent applications and travel grants".

6. We note that Figures 5 and 7 in your submission contain [map/satellite] images which may be copyrighted. All PLOS content is published under the Creative Commons Attribution License (CC BY 4.0), which means that the manuscript, images, and Supporting Information files will be freely available online, and any third party is permitted to access, download, copy, distribute, and use these materials in any way, even commercially, with proper attribution. For these reasons, we cannot publish previously copyrighted maps or satellite images created using proprietary data, such as Google software (Google Maps, Street View, and Earth). For more information, see our copyright guidelines: http://journals.plos.org/plosone/s/licenses-and-copyright.

     1. You may seek permission from the original copyright holder of Figures 5 and 7 to publish the content specifically under the CC BY 4.0 license.  

Reviewers' comments:

Reviewer's Responses to Questions

**Comments to the Author**

1. Is the manuscript technically sound, and do the data support the conclusions?

Reviewer #1: Yes

Reviewer #2: Yes

2. Has the statistical analysis been performed appropriately and rigorously? 

Reviewer #1: Yes

Reviewer #2: Yes

3. Have the authors made all data underlying the findings in their manuscript fully available?

Reviewer #1: Yes

Reviewer #2: Yes

4. Is the manuscript presented in an intelligible fashion and written in standard English?

Reviewer #1: Yes

Reviewer #2: Yes

5. Review Comments to the Author

Reviewer #1: PLOS ONE

PONE-D-23-29703 - [EMID:d1fc004c3e3e78de]

Title: Suppression of the alpha, delta, and omicron variants of SARS-Cov-2 in Taiwan.

General Comments:

This article shares Taiwan's experience in combating the Alpha and Delta virus strains, which have higher mortality rates despite the lack of a vaccine. In the face of the Omicron virus, which is highly contagious but has a relatively mild fatality rate, Taiwan has also gradually opened its borders and successfully conquered the Omicron epidemic. This article describes four key factors underlying Taiwan's success in controlling the spread of COVID-19: quick responses; effective control measures with new technologies and rolling knowledge updates; unity and cooperation among Taiwanese government agencies, private companies and organizations, and individual citizens; and Taiwanese self-discipline.

I think these experiences are worth sharing with other countries. The topic of this article falls within the scope of this journal. I have the following suggestions.

1. Taiwan’s reality is different from many other countries, in part because it only borders the sea. Nonetheless, the timeliness of border management measures (earlier introduction of entry restrictions and health screening) may have contributed to Taiwan's relatively low number of cases. This article does not explore this as the early introduction of border controls is mentioned but its impact is not discussed further or compared with other countries. I suggest the author compare the epidemic outcomes-related performance of Taiwan and other island countries. Please discuss whether island countries can help prevent the epidemic.

2. This article presented four factors for controlling the spread of COVID-19 in Taiwan. Nevertheless, the importance of mandatory interventions (contact tracing, quarantine, border control) versus non-mandatory interventions (use of masks, social distancing) was not discussed. Individual self-discipline may have played a smaller role than mandatory restrictions that can be backed-up by legal legislation. I suggest the author mention this in the discussion section. Some modelling studies may be added in the discussion, for example:

(1) The effect of preventing subclinical transmission on the containment of COVID-19: Mathematical modeling and experience in Taiwan. https://www.sciencedirect.com/science/article/pii/S1551714420301798

(2) Relative role of border restrictions, case finding and contact tracing in controlling SARS-CoV-2 in the presence of undetected transmission: a mathematical modelling study

https://www.ncbi.nlm.nih.gov/pmc/articles/PMC10019421/

3. A modelling analysis using Taiwan case data concluded that population-based interventions may have played a major role in Taiwan's initial elimination efforts and that case-based interventions alone were insufficient to control the epidemic. I suggest the authors add the findings in the discussion section.

(1) Comparison of Estimated Effectiveness of Case-Based and Population-Based Interventions on COVID-19 Containment in Taiwan. https://jamanetwork.com/journals/jamainternalmedicine/fullarticle/2778395

4. Several articles have discussed Taiwan’s experience and potential lessons in combating COVID-19, and this article does not add to what has been published. It would be important to cite and compare previous works on the Taiwanese experience, i.e., in order to support some of the results presented in the manuscript. For example,

(1) Comparison of Estimated Effectiveness of Case-Based and Population-Based Interventions on COVID-19 Containment in Taiwan. https://jamanetwork.com/journals/jamainternalmedicine/fullarticle/2778395

(2) Fighting COVID‐19 through government initiatives and collaborative governance: the Taiwan experience. Public Administration Review, 80(4), 665-670.

(3) An American's perspective living through COVID-19 in Taiwan. Journal of the Formosan Medical Association, 119(12), 1884.

5. Contact tracing is considered an important strategy to curb the spread of COVID-19. While this is true, no relevant references are provided to support this confirmation. It would be interesting to discuss the transmissibility of COVID-19) to close contacts. For example, there is a report from Taiwan on the use of contact tracing and testing to assess COVID-19 transmission dynamics among the first 100 confirmed cases in the country.

(1) Contact tracing assessment of COVID-19 transmission dynamics in Taiwan and risk at different exposure periods before and after symptom onset. JAMA internal medicine, 180(9), 1156-1163.

6. The findings underscore the need for accurate, comprehensive contact tracing and testing. However, they also stressed that simply identifying and isolating symptomatic patients may not be enough to interrupt transmission, and that social distancing measures will also be needed. A modeling study also suggests that isolating all at-risk cases through contact tracing can control the outbreak.

(1) The effect of preventing subclinical transmission on the containment of COVID-19: Mathematical modeling and experience in Taiwan. https://www.sciencedirect.com/science/article/pii/S1551714420301798

Reviewer #2: I read the manuscript “Suppression of the alpha, delta, and omicron variants of SARS-Cov-2 in Taiwan.” and I thank the authors and the journal for the possibility of reviewing this manuscript.

The authors describe Taiwan's responses to alpha, delta, and omicron variants. Taiwan has controlled its borders and implemented strict quarantine rules in the first two years of pandemic (2020 and 2021), keeping infection and deaths numbers low. Taiwan then gradually eased border measures in March 2023, when Taiwan faced a new variant of concern: Omicron. In this manuscript, four factors for controlling the spread of COVID-19 in Taiwan are presented. They are

(1) Quick response;

(2) Effective control measures and precise quarantine strategy;

(3) Individual self-discipline;

(4) Unity and cooperation.

The paper provides valuable perspective on Taiwan's success in controlling the spread of COVID-19, especially in the face of alpha and delta variants. I think this article can be improved by addressing the following issues. It would be beneficial for the authors to broaden their horizons and consider other infectious diseases that may cause pandemics in the future.

(1)The abstract should include some methods and results, not just the background and conclusion.

(2)The methods section lacks some important details that readers need to evaluate the methods used. For example, the authors analyzed the time to border closure and conducted one-sample test to compare between Taiwan and Non-Taiwan countries prior to vaccine introduction. It’s unclear how the time to border closure was calculated. Please provide more details.

(3)More information is needed about the data sources and how the time-varying reproduction number (Rt) was calculated, including the model used to estimate Rt and any assumptions made in the analysis. The authors are suggested to provide link to the data sources and the code used for calculating Rt in order to encourage reproducibility of their results.

(4)Taiwan adopted a COVID-zero approach in the first two years of pandemic (2020 and 2021), keeping infection and deaths low. I wonder how the COVID-zero approach affects the economy or social health in Taiwan.

(5)In the discussion section, please indicate some of the limitations of your study.

Addressing these concerns would strengthen the paper's argument and provide a more comprehensive understanding of Taiwan's approach to controlling the spread of COVID-19.

6. PLOS authors have the option to publish the peer review history of their article (what does this mean?). If published, this will include your full peer review and any attached files.

Reviewer #1: No

Reviewer #2: No

---

## [Author Response · Author response to Decision Letter 0]

18 Feb 2024

Point-by-point response to the academic editor and reviewer(s)

Dear Dr. Tsou,

Thank you for submitting your manuscript to PLOS ONE. After careful consideration, we feel that it has merit but does not fully meet PLOS ONE’s publication criteria as it currently stands. Therefore, we invite you to submit a revised version of the manuscript that addresses the points raised during the review process.

Thank you for submitting your important manuscript to PLOS ONE.

I think the comments raised by the 2 independent reviewers seem minor.

Please submit your revised version soon.

Authors’ reply:

Thanks very much for the Editor’s comments. Yes. We have paid careful attention to each of the comments that have been pasted underneath the email. We have tried our best to respond to the reviewer's comments and addressed them carefully in the revision.

Journal Requirements:

Authors’ reply:

Thank you so much for the Editor’s comments. We confirmed that our manuscript meets PLOS ONE's style requirements, including those for file naming.

2. Thank you for stating the following in your Competing Interests section: "All authors declare no conflicts of interest, including employment, consultancies, stock ownership, honoraria, paid expert testimony, patent applications and travel grants".

Authors’ reply:

Thank you so much for the Editor’s comments. We stated "The authors have declared that no competing interests exist." in our cover letter.

Authors’ reply:

Thank you so much for the Editor’s comments. We had provided the link to the code used for the analysis in the data availability section and relevant URLs within our revised cover letter. The minimal data set is available at the GitHub repository [https://github.com/ChiehCheng/COVID-19-in-Taiwan]. The population flow data referenced above is part of the minimal data set for this study. However, we are afraid that the population flow data could not be provided because we had a business contract with Far EasTone Telecommunications. Therefore, we provided the consulting webpage of Far EasTone Telecommunications to anyone who wants to request for more information. Anyone who has an interest in this data set could contact the following consulting webpage of Far EasTone Telecommunications for more information. 

[https://enterprise.fetnet.net/content/ebu/tw/product/IOT/IOT_L3/smart-city-big-data.html]

Authors’ reply:

Thank you so much for the Editor’s comments. We have an ORCID ID for the corresponding author in Editorial Manager on papers submitted.

ORCID ID：0000-0001-6773-4111

The corresponding author Hsiao-Hui Tsou had contacted Dr. Roxanne Jastine Baltar, a Straive Editorial Assistant on December 1, 2023 to knew that I am not able to retrieve my ORCID due to my multiple accounts in PLOS ONE system. So I am waiting for Dr. Roxanne Jastine Baltar’s help to merge my two accounts into one account (the one with username: tsouhh@nhri.edu.tw). Then I should be able to login into my account (the one with username: tsouhh@nhri.edu.tw) and retrieve my ORCID.

Authors’ reply:

Thank you so much for the Editor’s comments. We stated our full ethics statement in the ‘Methods’ section of our manuscript file.

*Lines 158-160 in “Methods”

6. We note that Figures 5 and 7 in your submission contain [map/satellite] images which may be copyrighted. All PLOS content is published under the Creative Commons Attribution License (CC BY 4.0), which means that the manuscript, images, and Supporting Information files will be freely available online, and any third party is permitted to access, download, copy, distribute, and use these materials in any way, even commercially, with proper attribution. For these reasons, we cannot publish previously copyrighted maps or satellite images created using proprietary data, such as Google software (Google Maps, Street View, and Earth). For more information, see our copyright guidelines: http://journals.plos.org/plosone/s/licenses-and-copyright.

 1. You may seek permission from the original copyright holder of Figures 5 and 7 to publish the content specifically under the CC BY 4.0 license. 

Authors’ reply:

Thank you very much for the Editor’s comments. Figure 5 in our submission is not copyrighted. Figure 5 was not created using proprietary data, such as Google software (Google Maps, Street View, and Earth). Instead, Figure 5 was created by an open shapefile which can be obtained from an open data platform (https://segis.moi.gov.tw/STAT/Web/Portal/STAT_PortalHome.aspx). Therefore, Figure 5 complies with the CC BY 4.0 license.

Figure 7 was not created using proprietary data, such as Google software (Google Maps, Street View, and Earth). Instead, Figure 7 was created by an open shapefile which can be obtained from an open data platform (https://odportal.tw/dataset/_5NxY3tq). 

our copyright: https://data.gov.tw/license.

Authors’ reply:

Thank you very much for the Editor’s comments. We have reviewed our reference list to ensure it is complete and correct. We removed reference 5 and 21 and replaced them with relevant current references.

Reviewer #1: PLOS ONE

PONE-D-23-29703 - [EMID:d1fc004c3e3e78de]

Title: Suppression of the alpha, delta, and omicron variants of SARS-Cov-2 in Taiwan.

General Comments:

This article shares Taiwan's experience in combating the Alpha and Delta virus strains, which have higher mortality rates despite the lack of a vaccine. In the face of the Omicron virus, which is highly contagious but has a relatively mild fatality rate, Taiwan has also gradually opened its borders and successfully conquered the Omicron epidemic. This article describes four key factors underlying Taiwan's success in controlling the spread of COVID-19: quick responses; effective control measures with new technologies and rolling knowledge updates; unity and cooperation among Taiwanese government agencies, private companies and organizations, and individual citizens; and Taiwanese self-discipline.

I think these experiences are worth sharing with other countries. The topic of this article falls within the scope of this journal. I have the following suggestions.

1. Taiwan’s reality is different from many other countries, in part because it only borders the sea. Nonetheless, the timeliness of border management measures (earlier introduction of entry restrictions and health screening) may have contributed to Taiwan's relatively low number of cases. This article does not explore this as the early introduction of border controls is mentioned but its impact is not discussed further or compared with other countries. I suggest the author compare the epidemic outcomes-related performance of Taiwan and other island countries. Please discuss whether island countries can help prevent the epidemic.

Authors’ reply:

Thank you so much for the reviewer’s comments. We compared the epidemic outcomes-related performance of Taiwan and other island countries and added this finding to the supporting information 1. 

 Appendix

supporting information 1.

“Figure S1 depicts the implementation of border control measures by island countries in response to the emergence of the first COVID-19 case in China, highlighting Taiwan's rapid deployment of border control measures. Following the implementation of border control measures, Taiwan recorded its first confirmed COVID-19 case after a 21-day interval. Despite the increase in the cumulative number of deaths per million people since May 2022, Taiwan's overall performance remains commendable when compared to other island countries, based on data from Our World in Data. (Figure S2)”

Supplementary Figure S1. The time to any border closure from first reported case in China and reference country.

Supplementary Figure S2. Cumulative deaths per million people in 2020-2022.

2. This article presented four factors for controlling the spread of COVID-19 in Taiwan. Nevertheless, the importance of mandatory interventions (contact tracing, quarantine, border control) versus non-mandatory interventions (use of masks, social distancing) was not discussed. Individual self-discipline may have played a smaller role than mandatory restrictions that can be backed-up by legal legislation. I suggest the author mention this in the discussion section. Some modelling studies may be added in the discussion, for example:

(1) The effect of preventing subclinical transmission on the containment of COVID-19: Mathematical modeling and experience in Taiwan. https://www.sciencedirect.com/science/article/pii/S1551714420301798

(2) Relative role of border restrictions, case finding and contact tracing in controlling SARS-CoV-2 in the presence of undetected transmission: a mathematical modelling study

https://www.ncbi.nlm.nih.gov/pmc/articles/PMC10019421/

Authors’ reply:

Thanks for the reviewer’s kind suggestion. Both case-based interventions (e.g., contact tracing, quarantine, border control) and population-based interventions (e.g., use of masks, social distancing) may be important to control the spread of COVID-19 in Taiwan. Although individual self-discipline could not be forced by legal legislation that mandatory restrictions could be, individual self-discipline would also be needed to interrupt transmission. We have mentioned this and added some modeling studies in the discussion section. 

 “Discussion”

“Both case-based interventions (e.g., contact tracing, quarantine, border control) and population-based interventions (e.g., use of masks, social distancing) may be important to control the spread of COVID-19 in Taiwan. It is necessary to not only have accurate, comprehensive contract tracing and testing, but also it is essential to identify and isolate symptomatic patients or all at-risk cases to control the outbreak [25, 50]. Although individual self-discipline is unable to be enforced by legal legislation while mandatory restrictions can, individual self-discipline would would reinforce the possibility of interrupting transmission. In other words, either category of interventions alone was insufficient to control the epidemic.”

*Lines 391-398 in “Discussion”

3. A modelling analysis using Taiwan case data concluded that population-based interventions may have played a major role in Taiwan's initial elimination efforts and that case-based interventions alone were insufficient to control the epidemic. I suggest the authors add the findings in the discussion section.

(1) Comparison of Estimated Effectiveness of Case-Based and Population-Based Interventions on COVID-19 Containment in Taiwan. https://jamanetwork.com/journals/jamainternalmedicine/fullarticle/2778395

Authors’ reply:

Thanks for the reviewer’s constructive suggestion. Both the population-based interventions (including social distancing and face masks) and case-based interventions (including contact tracing and quarantine) may have played a major role in Taiwan's initial elimination efforts, and either category of interventions alone was insufficient to control the epidemic. We have added this finding to the discussion. 

 “Discussion”

“Both case-based interventions (e.g., contact tracing, quarantine, border control) and population-based interventions (e.g., use of masks, social distancing) may be important to control the spread of COVID-19 in Taiwan. It is necessary to not only have accurate, comprehensive contract tracing and testing, but also it is essential to identify and isolate symptomatic patients or all at-risk cases to control the outbreak [25, 50]. Although individual self-discipline is unable to be enforced by legal legislation while mandatory restrictions can, individual self-discipline would would reinforce the possibility of interrupting transmission. In other words, either category of interventions alone was insufficient to control the epidemic.”

*Lines 391-398 in “Discussion”

4. Several articles have discussed Taiwan’s experience and potential lessons in combating COVID-19, and this article does not add to what has been published. It would be important to cite and compare previous works on the Taiwanese experience, i.e., in order to support some of the results presented in the manuscript. For example,

(1) Comparison of Estimated Effectiveness of Case-Based and Population-Based Interventions on COVID-19 Containment in Taiwan. https://jamanetwork.com/journals/jamainternalmedicine/fullarticle/2778395

(2) Fighting COVID‐19 through government initiatives and collaborative governance: the Taiwan experience. Public Administration Review, 80(4), 665-670.

(3) An American's perspective living through COVID-19 in Taiwan. Journal of the Formosan Medical Association, 119(12), 1884.

Authors’ reply:

We sincerely thank you for your review and valuable comments on our manuscript. We've mentioned and cited previous work in Taiwan as suggested. 

 “Discussion”

“The Taiwanese government setting operates the National Health Insurance (NHI) and Central Epidemic Command Center. The former ensures everyone has equal medical resources and helps the government check clinical records and allocate resources such as rapid tests and masks. The latter focues on the country's epidemic safety. At the individual level, they improved people's confidence in government and knowledge through transparent public press conferences and public health education videos. At the public level, border control and strict quarantine measures were promulgated very early, and case tracking and contact tracing were implemented effectively throughout the pandemic [20].”

*Lines 372-379 in “Discussion”

“Besides the government providing masks, Taiwan is “an island of masks” [49]. Different from the Western countries, sick people wearing masks in public is considered common and polite behavior to protect other people from being infected. This enhances the willingness and the number of people wearing masks.”

*Lines 383-386 in “Discussion”

“Taiwan’s control strategy includes the combination of case-based and population-based interventions which was indicated effective and may be interpreted as a reason for Taiwan's control success [15].”

*Lines 399-401 in “Discussion”

“Throughout the pandemic, the people of Taiwan have cooperated with public health laws and regulations and engaged in self-discipline. The common societal conformity led the Taiwanese to more immediately follow directives rather than hesitate and ask for the reason, which saves time during the epidemic [49], enabling NPIs to be highly effective and suppressing outbreaks due to emerging variants of concern.”

*Lines 405-409 in “Discussion”

5. Contact tracing is considered an important strategy to curb the spread of COVID-19. While this is true, no relevant references are provided to support this confirmation. It would be interesting to discuss the transmissibility of COVID-19) to close contacts. For example, there is a report from Taiwan on the use of contact tracing and testing to assess COVID-19 transmission dynamics among the first 100 confirmed cases in the country.

(1) Contact tracing assessment of COVID-19 transmission dynamics in Taiwan and risk at different exposure periods before and after symptom onset. JAMA internal medicine, 180(9), 1156-1163.

Authors’ reply:

Thank you very much for your valuable comments. We provided a relevant reference to support this confirmation. 

 “Factor 2”

This approach to disease control has several clear advantages. It not only provides a better understanding of the potential transmissibility of COVID-19, but also aids in the formulation of appropriate control strategies. Additionally, identifying the infection routes among healthcare workers enables the provision of suitable personal protective equipment (PPE) [23].

*Lines 225-229 in “Results”

6. The findings underscore the need for accurate, comprehensive contact tracing and testing. However, they also stressed that simply identifying and isolating symptomatic patients may not be enough to interrupt transmission, and that social distancing measures will also be needed. A modeling study also suggests that isolating all at-risk cases through contact tracing can control the outbreak.

(1) The effect of preventing subclinical transmission on the containment of COVID-19: Mathematical modeling and experience in Taiwan. https://www.sciencedirect.com/science/article/pii/S1551714420301798

Authors’ reply:

Thanks for the reviewer’s comment. It is necessary to not only have accurate, comprehensive contract tracing and testing, but also it is essential to identify and isolate symptomatic patients or all at-risk cases to control the outbreak. Furthermore, population-based interventions such as social distancing and face masks would also be needed to interrupt transmission. We have amended and added this citation to the discussion.

 “Discussion”

“Both case-based interventions (e.g., contact tracing, quarantine, border control) and population-based interventions (e.g., use of masks, social distancing) may be important to control the spread of COVID-19 in Taiwan. It is necessary to not only have accurate, comprehensive contract tracing and testing, but also it is essential to identify and isolate symptomatic patients or all at-risk cases to control the outbreak [25, 50]. Although individual self-discipline is unable to be enforced by legal legislation while mandatory restrictions can, individual self-discipline would would reinforce the possibility of interrupting transmission. In other words, either category of interventions alone was insufficient to control the epidemic.”

*Lines 391-398 in “Discussion”

Reviewer #2: I read the manuscript “Suppression of the alpha, delta, and omicron variants of SARS-Cov-2 in Taiwan.” and I thank the authors and the journal for the possibility of reviewing this manuscript.

The authors describe Taiwan's responses to alpha, delta, and omicron variants. Taiwan has controlled its borders and implemented strict quarantine rules in the first two years of pandemic (2020 and 2021), keeping infection and deaths numbers low. Taiwan then gradually eased border measures in March 2023, when Taiwan faced a new variant of concern: Omicron. In this manuscript, four factors for controlling the spread of COVID-19 in Taiwan are presented. They are

(1) Quick response;

(2) Effective control measures and precise quarantine strategy;

(3) Individual self-discipline;

(4) Unity and cooperation.

The paper provides valuable perspective on Taiwan's success in controlling the spread of COVID-19, especially in the face of alpha and delta variants. I think this article can be improved by addressing the following issues. It would be beneficial for the authors to broaden their horizons and consider other infectious diseases that may cause pandemics in the future.

(1)The abstract should include some methods and results, not just the background and conclusion.

Authors’ reply:

Thank you so much for the reviewer’s comment. We added methods and results in the abstract.

 Abstract

“ Background

Taiwan was a coronavirus disease 2019 (COVID-19) outlier, with an extraordinarily long transmission-free record: 253 days without locally transmitted infections while the rest of the world battled wave after wave of infection. The appearance of the alpha variant in May 2021, closely followed by the delta variant, disrupted this transmission-free streak. However, despite low vaccination coverage (<1%), outbreaks were well-controlled.

Methods

This study analyzed the time to border closure and conducted one-sample t-test to compare between Taiwan and Non-Taiwan countries prior to vaccine introduction. The study also collected case data to observe the dynamics of omicron transmission. Time-varying reproduction number, R_t, was calculated and was used to reflect infection impact at specified time points and model trends of future incidence.

Results

The study analyzed and compare the time to border closure in Taiwan and non-Taiwan countries. The mean times to any border closure from the first domestic case within each country were -21 and 5.98 days, respectively (P <.0001). The Taiwanese government invested in quick and effective contact tracing with a precise quarantine strategy in lieu of a strict lockdown. Residents followed recommendations based on self-discipline and unity. The self-discipline in action is evidenced in Google mobility reports. The central and local governments worked together to enact non-pharmaceutical interventions (NPIs), including universal masking, social distancing, limited unnecessary gatherings, systematic contact tracing, and enhanced quarantine measures. The people cooperated actively with pandemic-prevention regulations, including vaccination and preventive NPIs.

Conclusions

This article describes four key factors underlying Taiwan's success in controlling COVID-19 transmission: quick responses; effective control measures with new technologies and rolling knowledge updates; unity and cooperation among Taiwanese government agencies, private companies and organizations, and individual citizens; and Taiwanese self-discipline.”

*Lines 49-76 in “Abstract”

(2)The methods section lacks some important details that readers need to evaluate the methods used. For example, the authors analyzed the time to border closure and conducted one-sample test to compare between Taiwan and Non-Taiwan countries prior to vaccine introduction. It’s unclear how the time to border closure was calculated. Please provide more details.

Authors’ reply:

Thank you so much for the reviewer’s comment. Border control measures are determined using the Oxford COVID-19 Government Response Tracker (OxCGRT) indicator. We provide more details in the methods section.

 Methods

“We collected the variable of “international travel controls (C8)” from the Oxford dataset to determine the time (in days) between the first reported case to implementation of border closure in each country. We identified the initial time of policy implementation as the time when C8 first becomes nonzero. (The information of border control measures is in supporting information 1) Similarly, the first reported cases in each country are based on the COVID-19 dataset from Our World in Data. For example, the first reported case in China is 2019/12/31, border control measures in Taiwan were implemented on 2020/01/01, and the first case in Taiwan is 2021/01/22. The time to any border closure from first reported case in China and Taiwan are 1 and -21 day, respectively.”

*Lines 134-142 in “Methods”

 Appendix

Supporting Information 1. Border control measures

The Oxford COVID-19 Government Response Tracker (OxCGRT) gathers publicly accessible data on 24 indicators spanning government actions in the following four dimensions: (1) containment and closure, (2) economic response, (3) health systems, and (4) vaccine policies. Within the category of "containment and closure," specific indicators include the school closing (C1), workplace closing (C2), cancel public events (C3), restrictions on gatherings (C4), close public transport (C5), stay at home requirements (C6), restrictions on internal movement (C7), and international travel controls (C8). The international travel controls (C8) indicator is particularly used to assess border control measures. The coding for C8 is as follows: 0 - no restrictions, 1 - screening arrivals, 2 - quarantine arrivals from some or all regions, 3 - ban arrivals from some regions, 4 - ban on all regions or total border closure, and Blank - no data. This indicator provides valuable information on the extent of government measures in controlling international travel, contributing to a comprehensive understanding of a country's response to the COVID-19 pandemic.

(3)More information is needed about the data sources and how the time-varying reproduction number (Rt) was calculated, including the model used to estimate Rt and any assumptions made in the analysis. The authors are suggested to provide link to the data sources and the code used for calculating Rt in order to encourage reproducibility of their results.

Authors’ reply:

Thank you so much for the reviewer’s comments. The time-varying reproduction number R_t can be estimated using method of Cori et al. It is estimated by the ratio of number of new infections generated at time t. I_t is the total infectiousness of infected individuals at time t. w_s is a generation in interval or probability that s day separate the moment of infection in an index case and a daughter case. The formula of R_t by method of Cori et al. is 

R_t=I_t/(∑_(s=1)^t▒〖w_s I_(t-s) 〗).

R_t is also consider the average number of secondary cases that each infected individual would infect if the conditions remained as they were at time t. The corresponding package in R for evaluating Rt is called EpiEstim (https://cran.r-project.org/web/packages/EpiEstim/index.html). In this package, we used the function, called “estimate_R”, with the method of "parametric_si" to evaluate Rt. The information of serial interval is mean = 2.9 and sd = 1.5. The information of confirmed case is in file“221020 Number of confirmed cases.csv”. We have added this description to the supporting Information 2 of appendix .

(4)Taiwan adopted a COVID-zero approach in the first two years of pandemic (2020 and 2021), keeping infection and deaths low. I wonder how the COVID-zero approach affects the economy or social health in Taiwan.

Authors’ reply:

Thank you so much for the reviewer’s comments. We've added a description of how the COVID-zero approach affects the economy or social health in Taiwan in "Discussion"

 “Discussion”

“In examining the economic impact of COVID-zero approach in Taiwan in the first two years of pandemic (2020 and 2021), we focus on gross domestic product (GDP) and unemployment rate. Data on GDP and unemployment rate were collected from Trading Economics [51,52], while data on cumulative deaths per million people were collected from Our World in Data [11]. Acknowledging the emergence of the Omicron variant in late November 2021 [53], we limited our scope to the period before November 2021. Using the data available up to that point in time, we built scatter plots to illustrate the distribution of cumulative deaths per million people against GDP and unemployment rate. Figures of the supporting Information 9 illustrate that, in comparison to other countries, Taiwan’s cumulative deaths per million people and economic indicators such as unemployment rate between December 2020 and November 2021 are considered good since they are notably low (Supplementary Figure S13 and S14). Furthermore, an examination of the GDP data reveals that Taiwan's economy remained relatively stable in the first two years of pandemic (2020 and 2021), experiencing no significant fluctuations (Supplementary Figure S15 and S16).

During the initial stages of the COVID-19 outbreak, particularly in the Alpha and Delta phases, Taiwan distinguished itself for its exceptional performance and resilience across various metrics. According to the Mental Health Foundation's 2020 biannual national mental health index survey, Taiwan achieved an impressive overall score of 83.3[54]. Despite the challenges posed by the Omicron phase in 2023, Taiwan's 116th rank out of 194 countries points to a relatively low level of depression, with a depression rate of 3.59%. This rate is notably lower than that of Greece, which recorded the highest rate at 6.52% [55].

Furthermore, Taiwan's score of 83.8 on the 2023 country safety index by Numbeo solidifies its place as one of the top three safest countries, following only Qatar with a score of 85.7 and the United Arab Emirates with a score of 85.4 [56]. These achievements underscore the success of Taiwan's early Zero COVID-19 strategy. Importantly, the implementation of the COVID-zero approach during the initial two-year period (2020-2022) did not show any adverse psychological effects on the public's mental well-being or social psychological health, as evidenced by the survey findings [57].“

*Lines 410-437 in “ Discussion”

(5)In the discussion section, please indicate some of the limitations of your study.

Authors’ reply:

Thank you so much for the reviewer’s comments.We have added some of the limitations of our study in the discussion.

 “Discussion”

“This study has some limitations. First, it is commonly acknowledged that the susceptibility and transmissibility of the SARS-CoV-2 virus vary among countries. However, this study could not address interpersonal and geographical heterogeneity comprehensively, mainly because of limited data accessibility. This includes variations in predisposing conditions and compliance with epidemic control measures. Another constraint arises from the challenge of obtaining the vaccination status of those who are infected, making it challenging to assess the impact of vaccination on the Rt values.”

*Lines 438-444 in “Discussion”

---

## [Editor Report · Decision Letter 1]

26 Feb 2024

Suppression of the Alpha, Delta, and Omicron variants of SARS-Cov-2 in Taiwan.

PONE-D-23-29703R1

Dear Dr. Tsou,

We’re pleased to inform you that your manuscript has been judged scientifically suitable for publication and will be formally accepted for publication once it meets all outstanding technical requirements.

Kind regards,

Etsuro Ito, Ph.D.

Academic Editor

PLOS ONE

---

## [Editor Report · Acceptance letter]

5 Mar 2024

PONE-D-23-29703R1 

PLOS ONE

Dear Dr. Tsou, 

I'm pleased to inform you that your manuscript has been deemed suitable for publication in PLOS ONE. Congratulations! Your manuscript is now being handed over to our production team.

Kind regards, 

on behalf of

Prof. Etsuro Ito 

Academic Editor

PLOS ONE